Accelerated implementation for testing IID assumption of NIST SP 800-90B using GPU

Kim Yewon fdt150@kookmin.ac.kr 1
Yeom Yongjin 1 2
1 Department of Financial Information Security, Kookmin University , Seoul , South Korea
2 Department of Information Security, Cryptology, and Mathematics, Kookmin University , Seoul , South Korea
Mei Gang
Electronic publication date: 2021 Mar 8
Publication date: 2021
Volume: 7
Electronic Location ID: e404
Received 2020 Jun 2; Accepted 2021 Jan 30
Copyright: ©2021 Kim and Yeom
Copyright year: 2021
Copyright holder: Kim and Yeom
License: This is an open access article distributed under the terms of the Creative Commons Attribution License, which permits unrestricted use, distribution, reproduction and adaptation in any medium and for any purpose provided that it is properly attributed. For attribution, the original author(s), title, publication source (PeerJ Computer Science) and either DOI or URL of the article must be cited.
License URL: https://creativecommons.org/licenses/by/4.0/

Keywords: Parallel processing, GPU computing, Entropy estimator, NIST SP 800-90B, Random Number Generator

Funding: Institute for Information & Communications Technology Promotion (IITP) grant No. 2014-6-00908 This work was supported by an Institute for Information & Communications Technology Promotion (IITP) grant funded by the Korean Government (MSIT) (No. 2014-6-00908, Research on the Security of Random Number Generators and Embedded Devices). The funders had no role in study design, data collection and analysis, decision to publish, or preparation of the manuscript.

==============================
In cryptosystems and cryptographic modules, insufficient entropy of the noise sources that serve as the input into random number generator (RNG) may cause serious damage, such as compromising private keys. Therefore, it is necessary to estimate the entropy of the noise source as precisely as possible. The National Institute of Standards and Technology (NIST) published a standard document known as Special Publication (SP) 800-90B, which describes the method for estimating the entropy of the noise source that is the input into an RNG. The NIST offers two programs for running the entropy estimation process of SP 800-90B, which are written in Python and C++. The running time for estimating the entropy is more than one hour for each noise source. An RNG tends to use several noise sources in each operating system supported, and the noise sources are affected by the environment. Therefore, the NIST program should be run several times to analyze the security of RNG. The NIST estimation runtimes are a burden for developers as well as evaluators working for the Cryptographic Module Validation Program. In this study, we propose a GPU-based parallel implementation of the most time-consuming part of the entropy estimation, namely the independent and identically distributed (IID) assumption testing process. To achieve maximal GPU performance, we propose a scalable method that adjusts the optimal size of the global memory allocations depending on GPU capability and balances the workload between streaming multiprocessors. Our GPU-based implementation excluded one statistical test, which is not suitable for GPU implementation. We propose a hybrid CPU/GPU implementation that consists of our GPU-based program and the excluded statistical test that runs using OpenMP. The experimental results demonstrate that our method is about 3 to 25 times faster than that of the NIST package.

Introduction

A random number generator (RNG) generates random numbers required to construct the cryptographic keys, nonce, salt, and sensitive security parameters used in cryptosystems and cryptographic modules. In general, an RNG produces random numbers (output) via a deterministic algorithm, depending on the noise sources (input). If its input is affected by the low entropy of the noise sources, the output may be compromised. It is easy to find examples that show the importance of entropy in operating systems. Heninger et al. (2012) describes the RSA/DSA private keys for some TLS/SSH hosts may be obtained due to insufficient entropy of Linux pseudo-random number generator (PRNG) during the key generation process. Ding et al. (2014) investigated the amount of the entropy of Linux PRNG running on Android in boot-time. Kaplan et al. (2014) demonstrated an IPv6 denial of service attack and a stack canary bypass with the weaknesses of insufficient entropy in boot-time of Android. Kim, Han & Lee (2013) presented a technique to recover PreMasterSecret (PMS) of the first SSL session in Android by 258 complexity since PMS is generated from insufficient entropy of OpenSSL PRNG at boot-time. Ristenpart & Yilek (2010), Bernstein et al. (2013), Michaelis, Meyer & Schwenk (2013), Schneier et al. (2015), and Yoo, Kang & Yeom (2017) describe the attacks caused by weakness of entropy collectors or incorrect estimations of the entropy that are exaggerated or too conservative.

Insufficient entropy of the noise source that is the input into the RNG may cause serious damage in cryptosystems and cryptographic modules. Thus, it is necessary to estimate the entropy of the noise source as precisely as possible. The United States National Institute of Standards and Technology (NIST) Special Publication (SP) 800-90B (Barker & Kelsey, 2012; Sönmez Turan et al., 2016; Sönmez Turan et al., 2018) is a standard document for estimating the entropy of the noise source. The general flow of the entropy estimation process in SP 800-90B (Sönmez Turan et al., 2018) is to determine the track, estimate the entropy according to the track, and then apply the restart test, as summarized in Fig. 1. In this paper, determining the track is referred to as an independent and identically distributed (IID) test. There are two different tracks: an IID track and a non-IID track. If it is determined as the IID track, it is assumed that the samples of the noise source are IID; otherwise, the samples are non-IID. The estimator depending on IID or non-IID track estimates the entropy of the noise source. The restart test evaluates the estimated entropy using different outputs from many restarts of the noise source to check the overestimate. This document is currently used in the Cryptographic Module Validation Program (CMVP) and has been cited as a recommendation for entropy estimation in an ISO standard document ISO/IEC-20543 (2019) for test and analysis methods of RNGs. The principles of entropy estimators in SP 800-90B have been investigated and analyzed theoretically (Kang, Park & Yeom, 2017; Zhu et al., 2017; Zhu et al., 2019). However, it is difficult to find research on the efficient implementation of the entropy estimation process of SP 800-90B.

Figure 1 Flow of the entropy estimation process of SP 800-90B.

NIST provides two programs (NIST, 2015) on GitHub for the entropy estimation process of SP 800-90B. The first program is for the entropy estimation process of the second draft of SP 800-90B (Sönmez Turan et al., 2016), written in Python. The second program is for the entropy estimation process of the final version of SP 800-90B (Sönmez Turan et al., 2018), written in C++. Table 1 displays the execution times of two single-threaded NIST programs on the central processing unit (CPU). The noise source used as input is GetTickCount, with a sample size of 8 bits. GetTickCount can be collected through the GetTickCount() function in the Windows environment. Since GetTickCount is determined as the non-IID by the IID test, the process of the IID-track estimation entropy does not run. The entropy estimation process of the IID track takes approximately one second for both NIST programs if it is forcibly operated. In Table 1, the IID test consumes the majority of the total execution time in both programs.

Table 1 Execution time of each single-threaded NIST program for the entropy estimation process (noise source: GetTickCount; noise sample size: 8 bits).

	NIST program written in Python	NIST program written in C++	
IID test	17 h	1 h 10 min	
[IID track] Estimation entropy	−	−	
[Non-IID track] Estimation entropy	15 min	20 s	
Restart tests	2 s	2 min	
Total execution time	17 h 16 min	1 h 13 min	

Developers of cryptosystems or cryptographic modules should estimate the entropy of the noise sources to analyze the security of the RNG. Since the entropy estimation process of SP 800-90B is representative, and modules for the CMVP shall be tested for compliance with SP 800-90B (NIST & CSE, 2020), most developers use the method of SP 800-90B. Furthermore, since CMVP Implementation Guidance (IG) gives the link of the NIST programs (NIST & CSE, 2020), most developers use the NIST programs to reduce the time required for implementation. As recommended by the CMVP, the RNG should use at least one noise source. Since the NIST program estimates the entropy for one noise source, the developer should run the NIST program k times when the RNG uses k noise sources. Since the noise sources are different for each operating system, the developer should run the program k × s times if the developer’s cryptosystem or cryptographic module supports s operating systems. The distribution of the noise source may be changed due to mechanical or environmental changes or to the timing variations in human behavior (NIST & CSE, 2020). The physical noise source is based on a dedicated physical process (ISO/IEC-20543, 2019); it may be affected by the environment of the device in which the RNG operates. Therefore, to claim that the noise source has an identical distribution in any environment, the developer should perform the IID test and entropy estimation in several environments or devices. If the developer performs analysis on d devices, the developer should run the program k × s × d times. If k = 10, s = 2, and d = 5, the developer should run the NIST program 100 times. According to Table 1, the NIST program written in C++ requires approximately 1 h to estimate the entropy of one noise source. If the developer cannot run multiple NIST programs simultaneously, it takes about 100 hours or approximately four days. Moreover, to find k noise sources that can be used as inputs of the RNG in the environment, the developer should perform entropy estimation for k or more collectible noise sources. Therefore, it may take more than 100 hours. The developer of the cryptographic module for the CMVP should perform similar work for re-examination or new examination every specific period since the module will be placed on the CMVP active list for five years. The evaluator running checks based on the documentation submitted by the developer for the CMVP may run the NIST program multiple times as well. As this runtime may be burdensome for developers, it can be tempting to use an RNG without security analysis. Thus, if the developer’s RNG is vulnerable, this vulnerability is likely to affect the overall security of the cryptosystem or cryptographic module.

Graphics processing units (GPUs) are excellent candidates to accelerate the process of SP 800-90B, especially the IID test. GPUs were initially designed for accelerating computer graphics and image processing, but they have become more flexible, allowing them to be used for general computations in recent years. The use of GPUs for performing computations handled by CPUs is known as general-purpose computing on GPUs (GPGPUs). New parallel computing platforms and programming models, such as the computing unified device architecture (CUDA) released by NVIDIA, enable software developers to leverage GPGPUs for various applications. GPGPUs are used in cryptography as well as areas including signal processing and artificial intelligence. Numerous studies have been conducted on the parallel implementations of cryptographic algorithms such as AES, ECC, and RSA (Neves & Araujo, 2011; Li et al., 2012; Pan et al., 2016; Ma et al., 2017; Li et al., 2019) and on the acceleration of cryptanalysis, including hash collision attacks using GPUs (Stevens et al., 2017).

To process the entire IID test in parallel using GPU, approximately 9 GB or more of the global memory of the GPU are required. Since the compression test used in the IID test requires a different technique of implementation from the other statistical tests, a CUDA version of the compression test is needed to implement the IID test in parallel. However, bzip2 used in the compression test is not actively under development as a CUDA version since it is unsuitable for GPU implementation. Therefore, we propose a GPU-based parallel implementation of the IID test without the compression test using multiple optimization techniques. The adaptive size of the global memory used in the kernel function can be set so that maximal performance improvement can be obtained from the GPU specification in use. Moreover, we propose a hybrid CPU/GPU implementation of the IID test that includes the compression test. Our GPU-based implementation is approximately 12 times faster than the multi-threaded NIST program without the compression test when determining the noise source as the IID. It is approximately 25 times faster when determining the noise source as the non-IID. Our hybrid CPU/GPU implementation is 3 and 25 times, respectively, faster than the multi-threaded NIST program with the compression test when determining the noise source as the IID and the non-IID, respectively. Most noise sources are non-IID (Kelsey, 2012). The non-IID noise sources are disk timings, interrupt timings, jitter (Müller, 2020), GetTickCount, and so on. Since the proposed hybrid CPU/GPU implementation has better performance for the non-IID noise sources, we expect it to be highly practical.

The remainder of this paper is organized as follows. ‘Preliminaries’ introduces the CUDA GPU programming model, the OpenMP programming model, and the IID test of SP 800-90B. ‘Proposed Implementations’ outlines our GPU-based parallel implementation of the IID test and the hybrid CPU/GPU implementation of the IID test. In ‘Experiments and performance evaluation’, the experimental results on the optimization and performance of our methods are presented and analyzed. Finally, ‘Conclusions’ summarizes and concludes the paper.

Preliminaries

CUDA programming model

NVIDIA CUDA (NVIDIA, 2020b) is the most widely used programming model for GPUs. CUDA uses the single instruction multiple thread (SIMT) model. A kernel is a function that performs the same instruction on the GPU in parallel. A thread is the smallest unit operating the instructions of the kernel function. Multiple threads are grouped into a CUDA block, and multiple blocks are grouped into a grid.

A CUDA-capable GPU contains numerous CUDA cores, which are fundamental computing units and execute the threads. CUDA cores are collected into groups called streaming multiprocessors (SMs).

A kernel is launched from the host (CPU) to run on GPU and generate a collection of threads organized into blocks. Each CUDA block is assigned to one of the SMs on the GPU and executes independently on GPU. The mapping between blocks and SMs is done by a CUDA scheduler (Vaidya, 2018). An SM can concurrently execute the smaller group of threads, which is called a warp. All threads in a warp execute the same instruction, and there are 32 threads in a warp on most CUDA-capable GPUs. Latency can occur, such as data required for computation have not yet been fetched from global memory that the access is slow. To hide the latency, an SM can execute context-switching, which transfers control to another warp while waiting for the results.

The memory of CUDA-capable GPU includes global memory, local memory, shared memory, register, constant memory, and texture memory. Table 2 shows the memory types listed from top to bottom by access speed from fast to slow, and their principal characteristics.

Table 2 Memory of CUDA-capable GPU (NVIDIA, 2020a).

Memory	Location on/off chip	Access	Scope	Lifetime	
Register	On	R ∕W	1 thread	Thread	
Local	Off	R ∕W	1 thread	Thread	
Shared	On	R ∕W	All threads in block	Block	
Global	Off	R ∕W	All threads +host	Host allocation	
Constant	Off	R	All threads +host	Host allocation	
Texture	Off	R	All threads +host	Host allocation	

A basic frame of the program using the CUDA programming model is as follows: allocate memory in the device (GPU) and transfer data from the host to the device (if necessary); launch the kernel; transfer data from the device to the host (if required).

OpenMP programming model

Open Multi-Processing (OpenMP) (OpenMP, 2018) is an application programming interface (API) for parallel programming on the shared memory multiprocessors. It extends C, C++, and FORTRAN on many platforms, instruction-set architectures, and operating systems, including Linux and Windows with a set of compiler directives, library routines, and environment variables. OpenMP facilitates the parallelization of the sequential program. The programmer adds parallelization directives to loops or statements in the program.

OpenMP uses the fork-join parallelism (OpenMP, 2018). OpenMP program begins as a single thread of execution, called an initial thread. When the initial thread encounters a parallel construct, the thread spawns a team of itself and zero or more additional threads as needed and becomes the master of the new team. The statements and functions in the parallel region are executed in parallel by each thread in the team. All threads replicate the execution of the same code unless a work-sharing directive (such as for dividing the computation among threads) is specified within the parallel region. Variables default to shared among all threads in parallel region.

Terms

A sample is data obtained from one output of the (digitized) noise source and the sample size is the size of the (noise) sample in bits. For example, we collect a sample of the noise source GetTickCount in Windows by calling the GetTickCount() function once. In this case, the sample size is 32 bits. However, as certain estimators of SP 800-90B do not support samples larger than 8 bits, it is necessary to reduce the sample size. GetTickCount is the elapsed time (in milliseconds) since the system was started. Thus, it is thus easy to conclude that the low-order bits in the sample of GetTickCount contain most of the variability. Therefore, it would be reasonable to reduce the 32-bit sample to an 8-bit sample by using the lowest 8 bits. The entropy estimation of SP 800-90B is performed on input data consisting of one million samples, where each sample size is 8 bits. Furthermore, the maximum of the min-entropy per sample is 8.

IID test for entropy estimation

The IID test of SP 800-90B consists of permutation testing and five additional chi-square tests. Permutation testing identifies evidence against the null hypothesis that the noise source is IID. Since the permutation testing is the most time-consuming step in the entire IID test, we only focus on the permutation testing in this study.

_______________________ Algorithm 1 Permutation testing (S¨onmez Turan et al., 2018).___________________ Require: S = (s1,...,sL), where si is the noise sample and L = 1,000,000. Ensure: Decision on the IID assumption.   1:  for statistical test i do  2:     Assign the counters Ci,0 and Ci,1 to zero.   3:     Calculate the test statistic TEST INi on S.   4:  end for  5:  for j = 1 to 10,000 do  6:     Permute S using the Fisher–Yates shuffle algorithm.   7:     Calculate the test statistic TEST Shufflei on the shuffled data.   8:     if (TEST Shufflei > TEST INi) then  9:         Increment Ci,0. 10:     else if (TEST Shufflei = TEST INi) then 11:         Increment Ci,1. 12:     end if 13:  end for 14:  if ((Ci,0 + Ci,1 ≤ 5)or(Ci,0 ≥ 9,995)) for any i then 15:     Reject the IID assumption. 16:  else 17:     Assume that the noise source outputs are IID. 18:  end if______________________________________________________________________________

__________________________________________________________________________________________ Algorithm 2 Permutation testing of NIST program written in C++.____________ Require: S = (s1,...,sL), where si is the noise sample and L = 1,000,000. Ensure: Decision on the IID assumption.   1:  for statistical test i do  2:     Assign the counters Ci,0 and Ci,1 to zero.   3:     Calculate the test statistic TEST INi on S.   4:  end for  5:  for j = 1 to 10,000 do  6:     Permute S using the Fisher–Yates shuffle algorithm.   7:     for statistical test i do  8:         if statusi = true then  9:            Calculate the test statistic TEST Shufflei on the shuffled data. 10:            if (TEST Shufflei > TEST INi) then 11:                Increment Ci,0. 12:            else if (TEST Shufflei = TEST INi) then 13:                Increment Ci,1. 14:            else 15:                Increment Ci,2. 16:            end if 17:            if ((Ci,0 + Ci,1 > 5)and(Ci,1 + Ci,2 > 5)) then 18:                statei = false. 19:            end if 20:         end if 21:     end for 22:  end for 23:  if ((Ci,0 + Ci,1 ≤ 5)or(Ci,0 ≥ 9,995)) for any i then 24:     Reject the IID assumption. 25:  else 26:     Assume that the noise source outputs are IID. 27:  end if______________________________________________________________________________

__________________________________________________________________________________________ Algorithm 3 Fisher–Yates shuffle (S¨onmez Turan et al., 2018).___________________ Require: S = (s1,...,sL), where si is the noise sample and L = 1,000,000. Ensure: Shuffled S = (s1,...,sL).   1:  for i from L downto 1 do  2:     Generate a random integer j such that 1 ≤ j ≤ i.   3:     Swap sj and si.   4:  end for____________________________________________________________________________

Algorithm 1 presents the algorithm of the permutation testing described in SP 800-90B. The permutation testing first performs statistical tests on one million samples of the noise source, namely the original data. We refer to the results of the statistical tests as the original test statistics. Thereafter, permutation testing carries out 10, 000 iterations, as follows: In each iteration, the original data are shuffled, the statistical tests are performed on the shuffled data, and the results are compared with the original test statistics. After 10, 000 iterations, the ranking of the original test statistics among the shuffled test statistics is computed. If the rank belongs to the top 0.05% or bottom 0.05%, the permutation testing determines that the original data (input) are not IID. That is, it concludes that the original data are not IID if Eq. (1) is satisfied for any i that is the index of the statistical test. For any i, the counter Ci,0 is the number of j in step 5 of alg:alg1 satisfying the shuffled test statistic TESTiShuffle> the original test statistic TESTiIN. The counter Ci,1 is the number of j satisfying TESTiShuffle=TESTiIN, whereas the counter Ci,2 is the number of j satisfying TESTiShuffle<TESTiIN. (1) Ci,0+Ci,1≤5orCi,0≥9,995

Equivalently, the permutation testing determines that the original data are IID if Eq. (2) is satisfied for all i that is the index of the statistical test. (2) Ci,0+Ci,1>5andCi,1+Ci,2>5

The NIST optimized the permutation testing of the NIST program written in C++ using Eq. (2). Thus, even if each statistical test is not performed 10, 000 times completely, the permutation testing can determine that the input data are IID. Algorithm 2 is the improved version of the permutation testing optimized by the NIST.

We briefly introduce the shuffle algorithm and the tests used in the permutation testing. The shuffle algorithm is the Fisher–Yates shuffle algorithm presented in Algorithm 3. The permutation testing uses 11 statistical tests, the names of which are as follows:

• Excursion test

• Number of directional runs

• Length of directional runs

• Number of increases and decreases

• Number of runs based on the median

• Length of runs based on the median

• Average collision test statistic

• Maximum collision test statistic

• Periodicity test

• Covariance test

• Compression test*

The aim of the periodicity test is to measure the number of periodic structures in the input data. The aim of the covariance test is to measure the strength of the lagged correlation. Thus, the periodicity and covariance tests take a lag parameter as input and each test is repeated for five different values of the lag parameter: 1, 2, 8, 16, and 32 (Sönmez Turan et al., 2018). Therefore, a total of 19 statistical tests are used in the permutation testing.

If the input data are binary (that is, the sample size is 1 bit), one of two conversions is applied to the input data for some of the statistical tests. The descriptions of each conversion and the names of the statistical tests using that conversion are as follows (Sönmez Turan et al., 2018):

Conversion I

Conversion I divides the input data into 8-bit non-overlapping blocks and counts the number of 1s in each block. If the size of the final block is less than 8 bits, zeroes are appended. The numbers and lengths of directional runs, numbers of increases and decreases, periodicity test, and covariance test apply Conversion I to the input data.

Conversion II

Conversion II divides the input data into 8-bit non-overlapping blocks and calculates the integer value of each block. If the size of the final block is less than 8 bits, zeroes are appended. The average collision test statistic and maximum collision test statistic apply Conversion II to the input data.

For example, let the binary input data be (0, 1, 1, 0, 0, 1, 1, 0, 1, 0, 1, 1). For Conversion I, the first 8-bit block includes four 1s and the final block, which is not complete, includes three 1s. Thus, the output data of Conversion I are (4, 3). For Conversion II, the integer value of first block is 102 and the final block becomes (1, 0, 1, 1, 0, 0, 0, 0) with an integer value of 88. Thus, the output of Conversion II is (102, 88).

Proposed implementations

Target of GPU-based parallel processing

Steps 5 to 22 of Algorithm 2, with 10,000 iterations, consume most of the processing time of the permutation testing. The shuffle algorithm and 19 statistical tests are performed on the data with one million samples of the noise source in each iteration. Hence, it is natural to consider the GPU-based parallel implementation of 10,000 iterations, which are processed sequentially in the permutation testing.

The implementation of the compression test* differs from those of the other statistical tests used in the permutation testing. The compression test* uses bzip2 (Seward, 2019), which compresses the input data using the Burrows–Wheeler transform (BWT), the move-to-front (MTF) transform, and Huffman coding. There have been studies on the parallel implementation of bzip2 using the GPU. In Patel et al. (2012), all three main steps, namely the BWT, the MTF transform, and Huffman coding, were implemented in parallel using the GPU. However, the performance was 2.78 times slower than that of the CPU implementation. In Shastry et al. (2016), only the BWT was computed on the GPU and a performance improvement of 1.4 times that of the standard CPU-based algorithm was achieved. However, we couldn’t apply this approach, because our parallel test should be implemented on the GPU together with other statistical tests. Moreover, the compression test does not play a key role in Algorithm 2. That is, it is infrequent for a noise source to be determined as the non-IID only by the compression test results among the 19 statistical tests used in the permutation testing. Therefore, we design the GPU-based parallel implementation of the permutation testing consisting of the shuffle algorithm and 18 statistical tests, without the compression algorithm. Moreover, we design the hybrid CPU/GPU implementation of the permutation testing consisting of our GPU-based parallel implementation and a maximum of 10, 000 compression tests using OpenMP.

Overview of GPU-based parallel permutation testing

Approximately 9.3 GB (= 10, 000 × one million bytes of data) of the global memory of the GPU is required for the CPU to invoke a CUDA kernel to process 10, 000 iterations of the permutation testing in parallel on the GPU. Some GPUs do not have more than 9 GB of global memory. Therefore, we propose the GPU-based parallel implementation of the permutation testing, which processes N iterations in parallel on the GPU according to the user’s GPU specification and repeats this process R = ⌈10, 000∕N⌉ times.

Figure 2 CPU/GPU workflow of GPU-based parallel implementation of permutation testing.

(A) Code running on the host/CPU. (B) Code running on the device/GPU.

Figure 2 presents the workflow of the CPU and GPU. The host refers to a general CPU that executes the program sequentially, whereas the device refers to a parallel processor such as a GPU. In steps 1 to 3 of Fig. 2, the host performs 18 statistical tests on one million bytes of the input data (without shuffling) and holds the results. In step 4, the host calls a function that allocates the device memory required to process N iterations in parallel on the device. The use and size of the variables are listed in Table 3. In step 5, the input data (No. 1 in Table 3), and the results of the statistical tests in steps 1 to 3 (No. 4 in Table 3) are copied from the host to the device. In step 6, the host launches a CUDA kernel CurandInit, which initializes the N seeds used in the curand() function. The curand() function that generates random numbers using seeds on the device is invoked by the CUDA kernel Shuffling. When the host receives the completion of the kernel CurandInit, the host proceeds to steps 7 to 13.10, 000 iterations are divided into R rounds and each round processes N iterations in parallel on the device. To process N iterations, the host launches the CUDA kernel Shuffling (step 8) and then launches the CUDA kernel Statistical test (step 9) as soon as the host receives the completion of the kernel Shuffling. When the host receives the completion of the kernel Statistical test, in step 10, the counters Ci,0, Ci,1, and Ci,2 for i ∈ {1, 2, …, 18}, which indicate the indices of the statistical tests, are copied from the device to the host. Following the operations in steps 17 to 19 of Algorithm 2, which correspond to those in steps 12 and 13 of Fig. 2, the host moves on to step 14 if Eq. (2) is satisfied for all i. Finally, in step 14, the host determines whether or not the input data are IID.

Table 3 Use and size of variables allocated to GPU.

No.	Use of variable	Size of variable (bytes)	
1	Original data (input)	1, 000, 000	
2	N shuffled data	N × 1, 000, 000	
3	N seeds used by curand() function	N × sizeof(curandState) =N × 48	
4	18 Original test statistics	18 × sizeof(double) = 144	
5	Counter Ci,0, Ci,1, Ci,2 for 1 ≤ i ≤ 18	18 × sizeof(int) ×3 = 216	
6	N shuffled data after Conversion II (Only used if the input is binary)	N × 125, 000	

When the input data are binary, two conversions should be considered when designing the CUDA kernels. Therefore, we describe the CUDA kernels designed to process N iterations in parallel on the GPU depending on whether the input data are binary. The descriptions of the CUDA kernels Shuffling and Statistical test for non-binary noise sample are as follows:

CUDA kernel Shuffling

The kernel Shuffling generates N shuffled data by permuting one million bytes of the original data N times in parallel. Thus, each of N CUDA threads permutes the original data using the Fisher–Yates shuffle algorithm and then stores the shuffled data in the global memory of the device. As the shuffle algorithm uses the curand() function, each thread uses its unique seed that is initialized by the kernel CurandInit with its index, respectively.

CUDA kernel Statistical test

The kernel Statistical test performs 18 statistical tests on each of N shuffled data, and compares the shuffled and original test statistics. The size of each shuffled data is one million bytes and N shuffled data are stored in the global memory of the device. In this section, we present two methods that can easily be designed to handle this process in parallel on the GPU and propose an optimized method.

Parallelization method 1 One CUDA thread performs 18 statistical tests sequentially on one shuffled dataset. This method is illustrated in Fig. 3. If this method is applied to the kernel Statistical test, B′ = (N∕T) CUDA blocks are used when the number of CUDA threads is T. However, because each thread runs 18 tests in sequence, room for improvement is apparent in this method.

Parallelization method 2 In this method, each block performs its designated statistical test out of 18 tests on one shuffled dataset shared by 18 blocks. Thus, for one shuffled set, 18 statistical tests are run in parallel, and this method is a parallelization of the serial part in method 1 above. This method is illustrated in Fig. 4, which indicates the kernel Statistical test with B′ = ((N∕T) × 18) CUDA blocks and T threads in a block.

Proposed optimization This method optimizes parallelization method 2 through two steps. (Step 1) To hide the latency in accessing the slow global memory of the GPU, we analyzed the runtime of 18 statistical tests from an algorithmic perspective. We merged several statistical tests with similar access patterns to the global memory into a single test. Therefore, 9 merged statistical tests replace 18 statistical tests. (Step 2) When analyzed the execution time of nine merged tests, the execution time of one longest test was similar to the sum of the execution times of the remaining eight tests. We configured each thread of a block to runs the longest test and each thread of the other block to run eight merged tests so that the workload between SMs is balanced. This method is depicted in Fig. 5, where the kernel Statistical test uses B′ = ((N∕T) × 2) CUDA blocks, with T threads in each block.

With slight modifications to the kernels Shuffling and Statistical test, which are designed for non-binary samples, as described above, we can parallelize the permutation testing when the input data are binary. If the noise sample size is 1 bit, one of two conversions is applied to certain statistical tests. The data after Conversion I and data after Conversion II can be stored separately in the global memory. Since the data after Conversion I are the result of calculating the Hamming weight of the data following Conversion II, we designed to minimize the use of global memory as follows: In the kernel Shuffling, N CUDA threads first generate N shuffled data in parallel. Thereafter, each thread proceeds to Conversion II for its own shuffled data and stores the results (No. 6 in Table 3) in the global memory of the GPU. The kernel Statistical test runs nine merged tests. The merged tests that required Conversion I calculate the Hamming weight of the data after Conversion II. As in the optimized method for non-binary data, the thread in the block executes at least one test so that the execution time of each block is similar. Therefore, B′ = (N∕T) × 4 CUDA blocks are used when the number of CUDA threads is T.

Figure 3 General parallel method 1 of kernel Statistical test.

Figure 4 General parallel method 2 of kernel Statistical test.

Figure 5 Proposed optimization method of kernel Statistical test.

Overview of hybrid CPU/GPU implementation of permutation testing

We implemented the GPU-based permutation testing, which comprised 18 statistical tests without the compression algorithm and is parallel on the GPU. This section presents a hybrid CPU/GPU implementation of permutation testing that includes the compression algorithm.

As shown in Fig. 6, we designed the hybrid implementation to perform 10,000 shuffling and compression tests using OpenMP according to the result of our GPU-based permutation testing. The noise source is determined as the non-IID if at least one test does not satisfy Eq. (2), as shown in Algorithm 2. Therefore, if our GPU-based program determined that the input noise source is non-IID, our hybrid program finally determines that the input is non-IID, without compression tests. If our GPU-based program determined that the input is IID, the noise source might be determined to be IID or be determined to be non-IID only by the result of the compression test. Therefore, our hybrid program performs at most 10, 000 shuffling and compression tests in parallel using OpenMP. If the results of the compression tests satisfy Eq. (2), the noise source is finally determined as the IID; otherwise, it is determined as the non-IID.

Figure 6 Proposed hybrid CPU/GPU program of permutation testing.

(A) Process on the host/CPU. (B) Process on the device/GPU.

Experiments and performance evaluation

In this section, we analyze the performance of the proposed methods and compare its performance with the NIST program written in C++. The performance was evaluated using two hardware configurations (Table 4).

Table 4 Configurations of experimental platforms.

Name	Device A	Device B	
CPU model	Intel(R) Core (TM) i7-8086K	Intel(R) Core (TM) i7-7700	
CPU frequency	4.00 GHz	3.60 GHz	
CPU cores	6	4	
CPU threads	12	8	
Accelerator type	NVIDIA GPU	NVIDIA GPU	
Models	TITAN Xp	GeForce GTX 1060	
Multiprocessors (SMs)	30	10	
CUDA cores/SM	128	128	
CUDA capability major	6.1	6.1	
Global memory	12, 288 MB	6, 144 MB	
GPU Max clock rate	1, 582 MHz	1, 709 MHz	
Memory clock rate	5, 750 MHz	4, 004 MHz	
Registers/block	65, 536	65, 536	
Threads/SM	2, 048	2, 048	
Threads/block	1, 024	1, 024	
Warp size	32	32	
CUDA driver version	10.1	10.1	

Figure 7 Memory coalescing technique.

There are two noise sources used in experiments. The first noise source is truerand provided by the NIST. The second noise source, GetTickCount, could be collected through the GetTickCount() function in the Windows environment. The sample size of each noise source is 1, 4, or 8 bits. As a result of confirming whether the input data are IID by the IID test, truerand was determined as the IID noise source; however, GetTickCount was determined as the non-IID noise source.

The experimental result is the average of the results repeated 20 times. The difference between the results of the experiments repeated 20 times was within 5%. Since the GPU Boost technology, which controls the clock speed according to extra power availability, is used in NIVIDA GPU, the results are with the GPU Boost applied, unless otherwise noted.

GPU optimization concepts

We conducted experiments on the optimization concepts considered while GPU-based parallelizing the permutation testing. The experimental data used in this section consisted of one million samples collected from the noise source GetTickCount, where the sample size was 8 bits. In the experiments, we set T, the number of threads per block used in the CUDA kernel, to 256, a multiple of the warp size (=32). Since T is set to 256, we set N to 2, 048, which is the multiple T, and used about 2 GB (= N × 1, 000, 000 bytes) of the global memory of the GPU.

Coalesced memory access

We used the memory coalescing technique (Fig. 7) to transfer data from slow global memory to the registers efficiently. Table 5 displays the performance of our parallel implementation of the permutation testing before and after using this technique. Permutation testing used the kernel Statistical test with our optimization method. As a result, we improved performance by 1.5 times. All experiments after this section use the memory coalescing technology.

Merging statistical tests

Our optimization method consists of a step in which tests are merged (Step 1) and a step in which at least one test is allocated in the CUDA block so that the working time of each thread is similar (Step 2). Therefore, we confirmed the validity of our merged tests.

We first designed new CUDA kernels for experimentation, where each of the N threads performed one statistical test on one shuffled data. We measured the execution time of each test kernel. Each test kernel used eight CUDA blocks since we set the number of threads per block T to 256. The experimental results showing the execution time of each statistical test on the GPU are shown in Table 5.

From Table 6, it takes approximately four seconds if one thread sequentially performs 18 statistical tests. However, if one thread performs nine merged tests, it can be expected that it will take about 2.3 seconds. We improved the performance for all 18 statistical tests by about 1.7 times by combining the tests.

Table 5 Performance of proposed GPU-based parallel implementation of permutation testing depending on whether memory coalescing technique was used (the number of CUDA blocks = 16, the number of threads per block = 256).

	Before using memory coalescing technique (s)	After using memory coalescing technique (s)	
Device A	27.2	19.0	
Device B	54.1	33.9	

We measured the execution time of the parallelization method 2 applied Step 2, and our method. Referring to the results of Table 6, we designed each CUDA block of method 2 which Step 2 was applied to proceed with each of tests 1 ∼ 6, test 7, test 8, and tests 9 ∼ 18; each block can complete its work in a similar time. The kernel Statistical test applying this method uses 32 (=(N∕T) × 4) blocks; however, applying our proposed method uses 16 (=(N∕T) × 2) blocks. Table 7 presents the execution time of a kernel Statistical test with each method applied. As a result, our method is about 1.5 times faster than the parallelization method 2 applied Step 2.

Parallelism methods

We experimentally verified whether the proposed optimization method is better than other methods. We first confirmed the difference in the operation time of each CUDA thread in the kernel Statistical test, where each parallelization method is applied by drawing a figure. Figure 8 displays the operation times of the CUDA threads, assuming that the GPU had three SMs and considering the results of Table 6. It is the task of the GPU scheduler to allocate the CUDA blocks to the SMs; however, these were assigned arbitrarily for visualization in Fig. 8. As indicated in Table 6, the statistical tests had different execution times. Therefore, we expressed the different lengths of the threads in the CUDA blocks running each statistical test, as illustrated in Fig. 8. In the proposed method, several statistical tests were merged for optimization. The execution time of the merged statistical test (Table 6) was equal to or slightly longer than each execution time of the original statistical tests prior to merging (Table 6). Suppose that Test 1&2 is a merged function of Test 1 and Test 2. The lengths of the threads in the block running Test 1&2 were slightly longer than those of the threads in the block running Test 1 or Test 2, as indicated in Fig. 8. As illustrated in Fig. 8, we expected that our optimization outperformed parallelization methods 1 and 2.

Table 6 Left: execution time of each statistical test on GPU; right: execution time of each merged statistical test on GPU (Device A, number of CUDA blocks = 8, number of threads per block = 256).

No.	Name of statistical test	Execution time (ms)	No.	Name of merged statistical test	Execution time (ms)	
1	Excursion test	214	1′	Excursion test	214	
2	Number of directional runs	75	2′	Directional runs and number of inc/dec	90	
3	Length of directional runs	81				
4	Numbers of increases and decreases	38				
5	Number of runs based on median	103	3′	Runs based on median	143	
6	Length of runs based on median	128				
7	Average collision test statistic	1, 257	4′	Collision test statistic	1, 258	
8	Maximum collision test statistic	1, 238				
9	Periodicity test (lag = 1)	50	5′	Per/Cov test (lag = 1)	129	
10	Covariance test (lag = 1)	71				
11	Periodicity test (lag = 2)	94	6′	Per/Cov test (lag = 2)	137	
12	Covariance test (lag = 2)	113				
13	Periodicity test (lag = 8)	93	7′	Per/Cov test (lag = 8)	134	
14	Covariance test (lag = 8)	111				
15	Periodicity test (lag = 16)	93	8′	Per/Cov test (lag = 16)	134	
16	Covariance test (lag = 16)	111				
17	Periodicity test (lag = 32)	93	9′	Per/Cov test (lag = 32)	134	
18	Covariance test (lag = 32)	111				

Table 7 Performance of parallelization method 2 applied Step 2 and our method (Device A, the number of threads per block = 256).

	Number of CUDA blocks	Execution time (s)	
Parallelization method 2 (18 tests) +Step 2	32	2.24	
Our method (9 merged tests +Step 2)	16	1.51	

Figure 8 Operation times of CUDA threads in kernel Statistical test when applying each method on device.

We measured the execution time of a kernel Statistical test according to the parallel method. Table 8 shows the execution times of each kernel measured on both devices. If the occupancy of the kernel in our parallelization method is calculated, it reaches 100%. It is the occupancy per SM. Since our method uses a small number of blocks, there may be idle SMs on a high-performance GPU with many SMs. However, if the host calls the test kernel for each noise source simultaneously using a multi-stream technique, we can use almost full GPU capability.

Table 8 Execution time of kernel Statistical test according to parallel method (number of threads per block = 256).

		Execution time (s)	
Method	Number of CUDA blocks	Device A	Device B	
Parallelization method 1	8	4.53	6.39	
Parallelization method 2	144	2.77	6.33	
Our optimization (Step 1)	72	1.62	2.94	
Our optimization (Step 1&2)	16	1.51	2.76	

Since 18 statistical tests were running in parallel, the parallelization method 2 was improved by 1.6 times over method 1 in Device A; however, there was no improvement in the performance in Device B. In Device B, the number of SMs was 10, and the number of active blocks was calculated by eight. Thus, it is analyzed as the result derived since the number of blocks generated by the kernel (=144) is more than the number of blocks active in the device simultaneously (=80). Our method (Step 1) is about 1.7 and 2.1 times, respectively, faster than the parallelization method 2 in Device A and Device B. It is analyzed as the results due to the merged statistical tests that improved the performance, as confirmed in the previous section. Since the work of each CUDA block was adequately balanced, it is analyzed that our method (Step 1&2) was slightly improved over our method (Step 1). Furthermore, our method is 3 times and about 2.3 times, respectively, faster than the parallelization method 1 in Device A and Device B.

Next, we analyzed how each method affected the performance of GPU-based implementation of permutation testing. As shown in Algorithm 2, the permutation testing has 10, 000 iterations. Since implemented N iterations in parallel, the kernel CurandInit is called once, and the kernel Shuffling and Statistical test are called ⌈10, 000∕N⌉ times. Since we set N to 2, 048 and did not use Eq. (2) in this experiment, the permutation testing consists of one CurandInit, five Shuffling and five Statistical test. Figure 9 shows the execution time of this permutation testing according to the parallelization method. The permutation testing applied our method shows an improvement of about 1.8 times over the permutation testing applied method 1. Thus, our optimization method outperformed parallelization methods 1 and 2.

Figure 9 Execution time of the GPU-based parallel implementation of permutation testing according to parallel method (number of threads per block = 256).

Performance evaluation of GPU-based permutation testing according to the parameter

Parameter N is the number of iterations of the permutation testing to be processed in parallel. We measured the performance of the GPU-based parallel implementation of the permutation testing according to the value of the parameter N.

As shown in Fig. 2, the kernel CurandInit is called once. The kernel Shuffling and Statistical test are called at most ⌈10, 000∕N⌉ times. The calling process repeated is as follows: After the kernel Shuffling and the kernel Statistical test are sequentially run once, if the results do not satisfy Eq. (2), each kernel is called again. If each kernel has been called ⌈10, 000∕N⌉ times or the results satisfy Eq. (2), the call to each kernel is aborted.

If the noise source is IID, there is little evidence against the null hypothesis that the noise source is IID in the permutation testing. The probability of satisfying Eq. (2) increases, and the number of the calls of the kernel decreases. On the other hand, if the noise source is Non-IID, the probability of satisfying Eq. (2) decreases, and the number of the calls increases, contrary to the IID noise source case. Therefore, we used truerand and GetTickCount, which were determined as the IID and the non-IID, respectively, by permutation testing. The sample size of each noise source is 8 bits.

Permutation testing performs 10, 000 iterations, so we set N to be a factor of 10, 000 and T to 250. Since the size of the global memory in Device A is 12 GB, we set N to 1, 000, 2, 000, 2, 500, 5, 000, and 10, 000. In Device B, the size of the global memory is 6 GB, and so we set N to 1, 000, 2, 000, and 2, 500. Table 9 presents the execution time of the GPU-based parallel implementation of the permutation testing and the usage of global memory (calculated by referring to Table 3), according to the value N.

Table 9 Execution time of the GPU-based parallel permutation testing according to the value of the parameter N.

Parameter N	1, 000	2, 000	2, 500	5, 000	10, 000	
Global memory (GB)	0.93	1.86	2.33	4.66	9.31	
	Execution time (s)	
	truerand	2.69	3.78	4.53	9.20	19.76	
Device A	GetTickCount	26.92	18.81	18.19	18.43	19.83	
	truerand	3.59	6.80	8.58	−	−	
Device B	GetTickCount	35.75	33.97	34.49	−	−	

When truerand was used as input data, each of the kernel Shuffling and Statistical test was called once, and then the noise source was determined as the IID through the test results. Therefore, in an environment (e.g., Hardware RNG) where the noise sources are likely to be IID, it is analyzed that it is appropriate even if the user sets N to 1, 000. In GetTickCount, each kernel was called ⌈10, 000∕N⌉ times and then was determined as the non-IID. The execution time multiplied by ⌈10, 000∕N⌉, when truerand was the input, gives a similar result to the execution time when GetTickCount was the input. As shown in Table 9, in the case of GetTickCount, as N increases, the execution time decreases and then increases again. Each thread used the global memory of 1 million bytes. Therefore, we analyzed it as a result of the latency derived by increasing access to global memory as the number of switching by the warp unit increases. It is appropriate to select N by considering all of the global memory usages, execution time determined as an IID noise source, and execution time determined as a non-IID noise source in a general environment. As a result of the experiment, it is appropriate to set N to 2, 500 when using Device A and to select N to 2, 000 when using Device B.

Performance evaluation of GPU-based permutation testing with NIST program according to noise source

For each noise source, we measured the performances of our GPU-based program and the NIST program. Two noise sources, truerand and GetTickCount, were used in the experiment and the sample size of each noise source is one of 1, 4, and 8 bits. We set N to 2, 500 and 2, 000, respectively, when using Device A and Device B, reflecting the result of the previous experiment. We set T to 250.

The NIST program, written in C++, is compatible with OpenMP and can make 10, 000 iterations work in a multi-threaded environment. In this experiment, the NIST program running on the CPU used 12 CPU threads in Device A and eight CPU threads in Device B (Table 4). Thus, we compared our performance with permutation testing in the single-threaded and multi-threaded NIST programs. Since our GPU-based parallel implementation of the permutation testing was designed without the compression algorithm, we measured the performance of the NIST program without the compression test.

Table 10 presents the execution times of the NIST program on the CPU and the proposed program on the GPUs, measured for each noise source. For truerand, the performance of the proposed program was approximately 17.6 times better than that of the single-threaded NIST program. It was about 12.5 times better than the performance of the multi-threaded NIST program. In the case of GetTickCount, the performance of our program was improved by approximately 35.1 times and about 26.1 times over the single-threaded and the multi-threaded NIST programs.

Table 10 Performances of our GPU-based program and NIST program written in C++ according to noise source (without the compression test).

		Execution time (s)	
Name of noise source	truerand	GetTickCount	
	Sample size (bit)	1	4	8	1	4	8	
	NIST program (CPU single-thread)	43.42	77.52	24.94	434.42	485.58	638.89	
Device A	NIST program (CPU multi-thread)	37.53	54.91	23.66	331.76	339.79	347.68	
	Proposed program (GPU)	3.17	4.39	4.53	12.72	17.63	18.19	
	NIST program (CPU multi-thread)	41.35	50.15	23.18	361.23	347.15	353.52	
Device B	Proposed program (GPU)	4.60	5.91	6.80	23.01	29.58	33.97	

In Table 10, the minimum performance improvement of the proposed program for truerand was not higher than that of the program for GetTickCount. As shown in Algorithm 2, the number of iterations (up to 10,000) in permutation testing varies depending on whether Eq. (2) is satisfied. The NIST program on the CPU was executed as one statistical test unit. If the accumulated results of the statistical test satisfied Eq. (2), that test was no longer performed in the iterations. On the other hand, our program on the GPU was executed as an N unit of 18 statistical tests, and if the results of all tests satisfied Eq. (2), it was not repeated. Namely, the kernel Shuffling and Statistical test were not called again. If the noise source was likely to be determined as the IID from the permutation testing, there is a high probability that all of the statistical tests satisfy Eq. (2). The NIST program operating as one test unit repeatedly performed each test less than N times and then determined truerand as the IID; however, in the case of GetTickCount, both the NIST program and our program performed 10, 000 iterations and determined GetTickCount as the non-IID. Therefore, it is analyzed that the difference in performance improvement of our program by noise source is reasonable.

NVIDIA GPU Boost technology boosts the CUDA core frequency from 1, 582 to 1, 873 MHz in Device A. The execution time of our GPU-based program without GPU Boost is presented in Table 11. Without GPU Boost, the performance decreased by up to 0.96 times compared to the case with GPU Boost. It is analyzed that the difference in performance with or without GPU Boost is not significant. The performance of our GPU-based program without GPU Boost is approximately 5 to 34 times better than the single-threaded NIST program and about 5 to 25 times better than the multi-threaded NIST program.

Table 11 Execution time of the GPU-based parallel implementation of permutation testing with/without GPU Boost (Device A).

	Execution time (s)	
Name of noise source	With GPU Boost	Without GPU Boost	
truerand-1bit	3.17	3.21	
truerand-4bit	4.39	4.57	
truerand-8bit	4.53	4.66	
GetTickCount-1bit	12.72	12.87	
GetTickCount-4bit	17.63	18.28	

Performance evaluation of our hybrid CPU/GPU program

We measured the performance of the proposed hybrid CPU/GPU program and the NIST program using truerand and GetTickCount, whose sample size is 8 bits. Both programs included the compression test. Figure 10 presents the performance of each program. A base-10 logarithmic scale is used for the Y-axis.

Figure 10 Execution time of our hybrid program and NIST program.

Since the NIST program performs the compression tests, it takes longer than the runtime of the NIST program without the compression test written in Table 10. In particular, when determining GetTickCount to be non-IID, the compression test runs almost 10, 000 times, and so the NIST program, in this case, takes much longer than the runtime written in Table 10.

Our hybrid CPU/GPU program performs the compression tests using OpenMP only when our GPU-based program determined the noise source (e.g., truerand) as the IID. As shown in Fig. 10, it is reasonable that the execution time of our hybrid program for truerand is longer than that of our GPU-based program presented in Table 10. Since GetTickCount was determined as the non-IID by our GPU-based program, the compression test does not run in our hybrid program. Therefore, our hybrid program has the same execution time as our GPU-based program in Table 10.

Compared to the single-threaded NIST program, the proposed hybrid CPU/GPU program had an improved performance of approximately 4.9 to 192.9 times. Compared with the multi-threaded NIST program, the performance improved about 3.8 to 29.7 times. The NIST program always performed up to 10, 000 compression tests using OpenMP; however, our hybrid program performed the compression tests using OpenMP only if the noise source was determined as the IID by all 18 statistical tests in our GPU-based program. Therefore, our hybrid program is efficient when determining the noise source as the non-IID than when determining the noise source as the IID.

When the NIST program applies our implementation method, it first performs the shuffling and 18 statistical tests (at most 10, 000 times). If it determined that the noise source was non-IID by these results, it does not run the shuffling and the compression tests. When the input is non-IID, the NIST program (with the compression test) had the same runtime presented in Table 10. Otherwise, the NIST program has the same runtime as the original program. Therefore, our hybrid CPU/GPU program sped the process about 3 times over the multi-threaded NIST program applied our method for IID noise sources (8-bit sample size). Our program had an improved performance of approximately 25 for the non-IID input.

Conclusions

The security of modern cryptography is heavily reliant on sensitive security parameters such as encryption keys. RNGs should provide cryptosystems with ideal random bits, which are independent, unbiased, and, most importantly, unpredictable. To use a secure RNG, it is necessary to estimate its input entropy as precisely as possible. The NIST offers two programs for entropy estimations, as outlined in SP 800-90B. However, much time is required to manipulate several noise sources for an RNG.

We proposed GPU-based parallel implementation of the permutation testing, which required the longest execution time in the IID test of SP 800-90B. Our GPU-based implementation excluded the compression test that is unsuitable for CUDA version implementation. Our GPU-based method was designed to use massive parallelism of the GPU by balancing the execution time for statistical tests, as well as optimizing the use of the global memory for data shuffling. We experimentally compared our GPU optimization with the NIST program excluded the compression test. Our GPU-based program was approximately 3 to 34 times faster than the single-threaded NIST program. Moreover, our proposal improved the performance by about 3 to 25 times over the multi-threaded NIST program. We proposed the hybrid CPU/GPU implementation of the permutation testing. It consists of our GPU-based program and the compression tests that run using OpenMP. Experimental results show that the performance of our hybrid program is approximately 3 to 25 times better than that of the multi-threaded NIST program (with compression test). Most noise sources are non-IID, and our program has better performance when determining the noise source as the non-IID. It is expected that the time required for analyzing the RNG security will be significantly reduced for developers and evaluators by using the proposed approach, thereby improving the validation efficiency in the development of cryptographic modules. It is expected that our optimization techniques might be adapted to the problems of performing several tests or processes on thousands or more of data, each of which is large.

Additional Information and Declarations

Competing Interests

Author Contributions

Data Availability

The authors declare there are no competing interests.

Yewon Kim conceived and designed the experiments, performed the experiments, analyzed the data, performed the computation work, prepared figures and/or tables, and approved the final draft.

Yongjin Yeom conceived and designed the experiments, analyzed the data, authored or reviewed drafts of the paper, and approved the final draft.

The following information was supplied regarding data availability:

Data and source code are available at GitHub: https://github.com/yeah1kim/yeah_GPU_SP800_90B_IID.

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
