# Peer review of "Accelerated implementation for testing IID assumption of NIST SP 800-90B using GPU"

_PeerJ Computer Science, doi:10.7717/peerj-cs.404_

## Round 0.1 · original submission · Major Revisions

Please carefully read and respond to the reviewers' comments, especially the first and the second reviewers' comments for improving:
(1) the comparative experiments and (2) the implementation details.

·

Basic reporting

Paper is reasonably well structured, but the text is sometimes hard to read. I would suggest to seek the help of a native English speaker. If not possible, the general recommendation is to use shorter sentences which are easier to understand.

Experimental design

Authors propose a GPU implementation of subset of tests specified in NIST SP 800-90B. As I understand, the main performance bottleneck (in the straight-forward implementation) is utilization of GPU multi-processors. Only 8 CUDA blocks are scheduled if "Parallelization Method 1" is used. The authors propose to solve the problem by parallelizing across statistical tests - "Parallelization Method 2". They further suggests to combine several tests and execute them jointly to reduce load on GPU global memory. Authors suggest that 2x speed-up is achieved with these optimizations. I believe authors are right looking for performance in this 2 particular areas. But I think this is only first steps of the work and significantly higher performance gains are possible if authors take performance analysis and optimization in a more systematic way.

Even using the proposed optimization, many recent GPUs will be significantly under-utilized. With 72 blocks, "Device A" will be running at about 30% occupancy (64 warps at each of 30 SMs at maximum occupancy). Furthermore, according to table 5 "collision test statistic" takes order of magnitude more time than all other tests combined. Consequently, even more hardware resources will be idling while the 8 slowest blocks are computing collision statistics.
1. First of all, this aspect is not discussed in the paper. It should.
2. Then, there is a rather easy approach to increase occupancy if GPU memory allows it: just detect amount of available global memory and increase "N" (number of iterations) accordingly. Authors should discuss this option and explain why they decided against it.
3. It might be possible to further increase occupancy using a "parallel reduction" approach. Multiple threads may be assigned to compute a single test. It could be considerable work to optimize all tests, but it seems that majority of time is taken by "collision test" and it seems reasonable to invest time optimizing at least this one.

Validity of the findings

The performance-evaluation over-focus on work-to-thread mapping and register usage. This is only one, while important, aspect affecting the performance. From the presented analysis, it is even unclear if register usage play any considerable role. Even if they are, the included discussion is not rigorous enough. As I understand authors just present number of registers reported by NVIDIA compiler. This information can't be interpreted straightforward. CUDA optimizer generates code based on multiple constrains. Compiler often tries to reduce the number of used registers, but to do it the compiler might increase amount of computations (by re-computing some variables on the fly instead of assigning registers to them) or some variables might be spilled in the local memory. And this is only the simplest effect. There are more uses for registers. For instance, the registers are also used for streamlining access to the global memory. Even if amount of used registers have not increased, the compiler might have sacrificed something to allow higher occupancy. I don't think that it is happens in this particular case, but argumentation is not rigorous. Being not rigorous it does not contribute to the value of paper and should be rather omitted than presented in such form.

Additional comments

- The benchmarking procedure is not explained. Particularly, it would be interesting to learn if authors had considered effects of GPU boost while measuring performance. The clock rates of NVIDIA GPUs vary significantly depending on load and chip temperature and might introduce significant fluctuations in the measurements if not handled with care.

- The execution time in Table 5 sums up to 10.22. However, Table 4 gives only 9.38 for "Device A". Yes, I understand that fluctuations happen, but this is almost 10%. The same speed-up is claimed in the section "Coalesced memory access". How authors can be sure that it is a real effect of optimization and not another fluctuation in measurements?

- While comparing with CPU-version, it is interesting to see how optimal is the CPU version. Can we tune a CPU version instead of including GPU and get the same result? I.e. is it multi-threaded or single-threaded? Are performance optimizations considered in the code or not, e.g. are SIMD instructions used?

- The summary is a bit overstates the achieved results: "When applied to an IID noise source, the proposed program was 10 times faster than the NIST program written in C++. Moreover, for a non-IID noise source, our proposal improved the performance up to 23 times." According to table 7, 10 times was only achieved for sample sizes of 1- and 4-bits. The performance was only twice faster for the 8-bit use-case which was mostly discussed in the paper. Actually, it would be interesting to learn why the GPU performance favors 1- and 4-bit sample sizes. Considering GPU architecture, I would rather expect the opposite outcome.

- In the introduction, major motivation for work is that IID test needs to be executed repeatedly and, for this reason, performance is crucial (p.2 l. 73). However, as I understand once source is determined to be non-IID no further tests are needed. In the conclusion, the authors focus on speed-up for non-IID use case. But considering the Introduction this speed-up is relatively unimportant. The practical use-case is for IID tests. I think this should be made clear in the conclusion.

- Authors does not explain how bzip2 test was handled when the performance was compared between CPU and GPU implementations. There is no GPU version for this test. Have the GPU implementation executed 1-test less than the CPU? Or was the CPU-based bzip2 test executed in GPU-implementation? Or was bzip2 test excluded from both CPU and GPU implementations?

- Furthermore, it would be interesting how much time is required for bzip2 test compared to all other tests as presented in Table 5.

- On Fig.2 (and some others), letter "T" is used to denote 1) number of threads in block 2) result of statistics calculation. This is a bit misleading. I suggest to use different letters.

- In captions of Table 4 and 5, the authors state "number of CUDA blocks = 8". I guess this is a typo. Earlier authors say that 8 blocks are used for "Parallelization Method 1", but 72 for the optimized version (and 144 for method 2).

- p.4 l.141: typo "10,0000" => "10,000"

Reviewer 2 ·

Basic reporting

The paper proposes a GPU-based parallel implementation of the NIST SP 800-90B IID assumption testing, and carefully designs the implementation logic and finely control the resources. The proposed implementation shows several times speedup than the NIST package.
1) Overall, the paper is well written and easy to understand;
2) The introduction and background are concise, and give a clear mention of SP800-90B;
3) The presentation of algorithm and optimizations are good and ordered in a comprehensible fashion,
4) Figures and tables are comprehensive.

Here are some points that should be improved:
1) SP 800-90B relevant background is sufficient, but the preliminary of GPU and CUDA programming are absent that may impact on its readability for readers that are unfamiliar with GPU;
2) There are plenty of prior GPU-based cryptography relevant literatures, which should be appropriately referenced and reviewed. Some of the prior literatures should be updated to the state-of-the-art ones, e.g., Manavski, 2007; Szerwinski and Gu¨neysu, 2008;
3) The authors should explain the motivation further. The authors say that “the NIST program requires approximately 1 h to estimate the entropy of one noise source. Therefore, at least 20 h are required to analyze (20 noise sources)” (1st paragraph, Page 3). Why must the 20 tests execute serially? How about running the 20 tests in a 10-core hyperthreading CPU? Besides, as far as I know, SP 800-90B is a professional test mainly used by CMVP and its accredited labs, and only applied several times for each type of entropy. And it is an off-line test and rarely used in run-time of an RNG. All the above reasons make performance not a vital concern for SP 800-90B.
4) The basic situation of the NIST package should be clarified. “Written in C++/Python” is just not enough. Extra information should be introduced to reflect implementation level of NIST package, e.g., AVX acceleration, multi-thread feature etc. 23 times speedup sounds fine, but it is based on how good the reference is.

Experimental design

The basic idea of the experimental design is straight and clear, but it can be improved in two aspects:
1) the paper presents several optimization techniques, e.g., statistical tests merging and Memory coalescing. But there is no incremental analysis for the proposed solution to demonstrate how each optimization impacts the overall performance of the proposed solution.
2) the paper only compares the results with the CPU-based NIST package, but lacks of comparisons with relevant GPU-based applications. I cannot tell whether it is a high-level GPU-accelerated application.

Validity of the findings

It is a common sense that GPU-accelerated applications have great performance advantage over the typical CPU implementations. However, the performance advantage of this paper is not clear.
1) The authors claim “the experimental results demonstrate that our method is ‘at least’ 23 times faster than that of the NIST package”. However, Table 7 reports “up to” 22.5 times speedup than the NIST package.
2) And the paper does not report the experiment setups for the NIST programs, e.g., single-thread, or multi-thread, etc. According to Table 7, performance advantage is 2 to 23 times, in most of cases, 10 times. However, if the such advantage is achieved by comparing with single-thread NIST package, the result is disappointing.

Additional comments

I thank you for providing the source code, but I wonder whether the if-else semantics in the source code can run as expected under the SIMT architecture, i.e., the different statistical tests can really run concurrently as shown in Figure 7. I think the multi-stream technique may be more appropriate.

Reviewer 3 ·

Basic reporting

The paper mostly explains the general executions of the prior NIST algorithm on GPGPU platform. It is obvious that the parallel platform like GPGPU will enhance the execution speed of NIST considerably. However, the contribution of this research work to improve the NIST algorithm is very significant. At the same time, Coalesced memory access technique is a very small contribution in this work. Furthermore, the structuring of this paper is very confusing. Hence, the paper needs to be improved with further contributions and it requires complete restructuring.

Experimental design

The shown experimental design in this paper is not very innovative.

Validity of the findings

The findings are good but does not meet the expections.

Additional comments

Please try to improve the existing algorithm for claiming your contribution.

---

## Round 0.2 · Major Revisions

Please further improve your manuscript by specifically focusing on the following issues:

1) There are some problems with readability
2) Motivation for the work is not clear enough
3) "Conclusion" section overstates the results

·

Basic reporting

Authors have improved the paper and satisfactory answered a part of comments. Still I think
1) There are some problems with readability
2) Motivation for the work is not clear enough
3) "Conclusion" section overstates the results

I think the readers will benefit if authors present a real-world use-case for their application. Particularly the following aspects should be highlighted.
1) How often the sources already determined to be IID and Non-IID should be retested.
2) What is expected proportion between IID and Non-IID tests?
3) What is the estimate of noise sources which would be tested simultaneously?
4) In which environment this tests will be executed? A lab checking for compliance? Super-computing center? End-users?
I.e. a real example proving why the performance is critical.

Experimental design

The work is missing by far the slowest component of the test. In the rebuttal authors tell that the missing bzip2-test takes 40 minutes on CPU while other 18 implemented tests only 9 minutes. Then, authors claim that bzip2-test is not important and can be easily omitted.

This even might be true, but the authors talk about the formal test defined by formal specification. Without bzip2 test the proposed software is not complaint with NIST standard and probably can't be used in practice (or the practical use of incomplete test should be explained in the introduction).

Furthermore, I am not aware about active attempts to make CUDA-based implementation of bzip2 algorithm and I am highly skeptical about prospect of such implementation. So while the work have merits in solving part of the problem, the authors should clearly state in the abstract/conclusion that only part of the test was optimized. And that the slowest part of the test was excluded from the presented study due to complexity. Authors might elaborate if they have some expectations on GPU-version of bzip2. Otherwise, they also must make it clear that fast CUDA version of the bzip2 is not under development and the full version of the test might never materialize.

Furthermore, the official code might be pretty inefficient. Authors report very litle difference between single-threaded and parallel (Open-MP) implementation. If properly parallelized and optimized, the existing C implementation might give a comparable performance to the presented GPU version. Considering the situation with bzip2 test, this might be a better way forward if the performance of IID test is critical.

Validity of the findings

I. The authors added table which indicates number of used registers as reported by NVIDIA Nsight. This, however, doesn't answer the point raised in the original review. In the reviewed paper a full page is used to count registers generated by NVIDIA compiler and make trivial occupancy analysis. The number of the generated registers alone (without further analysis) means very little performance-wise. So, there is no practical value in this analysis.

In other words: Authors propose a different method to distribute work between threads. They have measured performance increase. This is a valid point. The following occupancy analysis doesn't contribute to the work because
1) It might be possible to achieve performance improvement even if more than 32-registers would be required and the occupancy would be lower.
2) The proposed optimization also might result in performance degradation even if occupancy stayed the same.

Registers and occupancy play a significant role, but it is more complex than "higher occupancy is always better". The analysis is misleading, does not contribute anything to the paper, and should be removed or significantly extended.

II. My comment about GPU boost was also not handled properly. NVIDIA GPUs modulates the clocks according to the load and temperature. This modulation can't be treated as random Gaussian noise and unlikely can be compensated by averaging 20 executions. A better approach is required to avoid unpredictable fluctuations of performance which e.g. on Pascal architecture my reach 20%.

If the GPUBoost can't be disabled on particular GPUs, the authors should either properly quantify the effect on the measurements or/and try to mitigate it. If the work-load fully loads GPU, one way is to wait until GPU temperature reaches GPUBoost maximum threshold and only, then, start real performance measurements. Then, the GPU will reset clocks to the minimum to avoid overheating and the clocks would likely not fluctuate further significantly. In any case, it is important to monitor the GPU clock rate during the benchmark to ensure that all tests are executed under the same conditions.

III. The performance presented in Table 7 (formerly Table 4) completely changed in the reviewed paper (also performance of the not optimized parallelization methods 1 & 2 has changed). Also, proportions have changed. E.g. in the original paper parallelization method 2 was 10% slower than method 1. But in the revised paper it is insignificantly faster. This makes me doubt claimed 5% of measurements volatility. And overall, the authors should explain to the reviewers the change in the measurement procedure or why all timings are now different.

IV. In p. 18, ll. 505... the authors try to find optimal number of N (number of test iterations performed in parallel). Generally, the higher amount of parallelism should result in the better performance (unless you can avoid part of processing like in IID case). But authors claim the best performance is achieved with N=2048 and it decreases with larger values of N. I think it requires explanation of causes, not just a mere constatation. I also guess that the sampled values of N were taken not optimally. Authors want to perform 10,000 iterations. So, ideally 10,000 should be multiple of N. Of course, also the number of scheduled blocks ideally should be proportional to the number of SMs on the device. So, it might make a sense to consider holistically the GPU architecture (number of SMs and amount of available memory), N, and blocking/work-allocation problem.

V. In the conclusion the authors still overstate the achieved performance improvements. The stated ratio of "33" is the maximum achieved in one test scenario. In other tests the performance ratio is between 4 and 33.

Additional comments

p. 11, l. 304-308. I don't understand explanation of Step2. It would be nice to reformulate


Overall, to accept for publications, I think authors must
1) Provide a good use-case scenario
2) Remove discussion about the number of registers or provide appropriate ptx analysis to prove it relevant.
3) Either account for GPUBoost in benchmarks or give a valid estimate of performance fluctuations. 5% is a clear underestimate.
4) The absence of bzip2 test should be clearly stated in the abstract and conclusion. The achieved performance should not be exaggerated in the conclusion.

Reviewer 2 ·

Basic reporting

no comment.

Experimental design

no comment.

Validity of the findings

no comment.

Additional comments

no comment.

---

## Round 0.3 · Minor Revisions

Please slightly improve your figures and tables according to the reviewer's comments.

·

Basic reporting

The authors made significant improvements over 2 iterations. They have further optimized their method and added many valuable details to the paper. The structure is clear and figures/explanations are well presented.

Authors addressed my comments and explained their vision on applications of the developed method. I am still have one question. I am not expert on cryptosystems and I am curious about the following: Lets say we identified some source as non-IID at Windows operating system. Do we really need to recheck it also on Linux and other operating systems? Can it really be IID in Windows and non-IID in Linux? Can authors reference an example?

Experimental design

I am not convinced that the official multi-threaded NIST code is optimal as presented in rebuttal letter. The authors provide the following run-times for 8-bit GetTickCount test:
- Table 10 in the previous revision of the paper:
638s for single-threaded execution without compression
347s for multi-threaded execution without compression
- Figure 10 in the current revision of the paper:
3510s for single-threaded execution with compression
541s for multi-threaded execution with compression

From these numbers I conclude that the compression test had significantly benefited from the multi-threaded execution, but less than 2x speed-up was achieved for all other tests. Consequently, I am still not completely sure that the selected approach is optimal for the goal. Instead I would rather recommend to improve CPU version. It should be possible to run compression test only conditionally as they do in the presented GPU-based approach. Considering little difference between single-threaded and parallel (Open-MP) implementations for the rest of the tests, I believe there is a significant space for improvements and a performance comparable to the presented by the authors could be achieved as well without GPUs. This would make adoption of a new method much easier.

Nevertheless, the work definitively have merits and describes an interesting optimization approach which might be used for other tasks. In a minor revision, I would suggest authors to summarize optimization strategy and suggest how this can be adapted to different problems. This is not prerequisite for publication from my point of view, but I believe it might contribute to the scientific value of the paper.

Validity of the findings

I am generally satisfied with with additions authors provided to handle GPU boost technology. However, the motivation in the following statement is not clear and might be incorrect: “It is analyzed that the difference in performance with or without GPU Boost is not significant because there are more integer operations than floating-point operations.”. Similarly to floating-point operations, execution of integer operations depend on the current clock. So, it is unclear why it should make any difference (the actual reason could be actually that the code is mostly memory bound and as I can see from the image attached to rebuttal letter the memory clock is not scaled with GPU boost). Anyway, authors either need to clarify this or remove the last part of the sentence. For the presented performance evaluation, it is important that GPU-boost doesn’t affect results much. The analysis why it is the case might be considered out of the scope of the paper.

Additional comments

I would suggest to improve Figure 10. It seems to use logarithmic Y-scale. However, the equidistant horizontal grid lines make an illusion that the scale is linear. I will suggest to make it more apparent that the Y-axis is logarithmic either by labeling the Y-axis or making grid lines non equidistant.

---

## Round 0.4 · accepted · Accept

The manuscript has been well revised. I think it is ready for acceptance. Before submitting your final files, please enlarge the font size in Figure 6 and Figure 8. Currently, the font size is too small.

·

Basic reporting

My comments were addressed and I think the paper deserves to be published.

Experimental design

-

Validity of the findings

-

Additional comments

-